# Association between the Number of Prosthetic Crowns and Periodontitis: The Korea National Health and Nutrition Examination Survey (KNANES VII) from 2016–2018

**DOI:** 10.3390/ijerph18115957

**Published:** 2021-06-02

**Authors:** Yun-Jeong Kim, Jae-Young Lee, Young Ku, Hyun-Jae Cho

**Affiliations:** 1Department of Periodontology, Seoul National University Gwanak Dental Hospital, Seoul 08826, Korea; yjbest01@gmail.com; 2Department of Dental Hygiene, College of Health Science, Dankook University, Cheonan 31116, Korea; dentaljy@dankook.ac.kr; 3Department of Periodontology, School of Dentistry, Seoul National University, Seoul 03080, Korea; guy@snu.ac.kr; 4Dental Research Institute, Seoul National University, Seoul 03080, Korea; 5Department of Preventive Dentistry and Public Oral Health, School of Dentistry, Seoul National University, Seoul 03080, Korea

**Keywords:** cross-sectional study, periodontitis, prosthetic crowns, oral health, oral hygiene, prevention

## Abstract

Although the presence of prosthetic restorations has been associated with plaque accumulation, gingivitis, and periodontitis, there is a lack of large epidemiological investigations providing evidence on the association of prosthetic crowns with periodontitis. This study aimed to analyze the association between the number of prosthetic crowns and the presence of periodontitis. This study was based on the Seventh Korea National Health and Nutrition Examination Survey (2016–2018). A total of 12,689 participants over the age of 19 years were surveyed. Multivariate logistic regression analyses were used to identify the association between the number of prosthetic crowns and periodontitis after adjusting for potential confounders, including demographic variables, socio-economic characteristics, oral health-related variables, and oral and systemic clinical variables. The odds ratio of periodontitis showed statistically significant differences in the anterior and posterior regions, and the prevalence of periodontal disease increased as the number of crown prostheses increased. Participants with 6–10 and 11 prosthetic crowns had 1.24 and 1.28 times higher prevalence of periodontitis, respectively, than patients with no prosthetic crown. The results of this study show that the number of prosthetic crowns present in adults is related to the prevalence of periodontitis.

## 1. Introduction

Periodontitis refers to host-mediated inflammation associated with a bacterial biofilm that causes the loss of periodontal attachment [1]. The initiation and progression of periodontitis are dependent on the ecological changes caused by the dysbiosis of microorganisms that develops as anti-bacterial mechanisms generating inflammation and tissue destruction in the gingival sulcus [2]. As periodontitis progresses, there is crestal bone resorption along with loss of attachment caused by inflammatory mediators, and the depth of the periodontal pocket increases. 

The presence of prostheses restored on the teeth, along with the anatomy, position, and the relationships of the teeth with each other is associated with plaque accumulation and subsequent gingivitis and periodontitis [3]. The process of fabrication and delivery of tooth-supported prostheses also has the potential to affect periodontal tissues by causing trauma or due to allergic reactions to dental materials [4]. It has been reported that the deeper the placement of the subgingival crown margin in the gingival sulcus, the more severe the gingival inflammation [5]. Additionally, long-term observations have shown that loss of attachment rapidly progresses between one and three years after placing subgingival restorations [6]. Other studies have reported that the subgingival margin design of prostheses affects the adjacent periodontal tissue in terms of bleeding on probing and gingival recession [7]. It is also widely known that overhanging restorations cause gingival inflammation and interproximal bone loss [8,9]. These observations are derived from the premise that it is not easy to control the plaque around the subgingival margin and the overhanging prosthesis.

Nevertheless, to date, there are no large epidemiological investigations providing evidence on the association of prosthetic crowns with periodontitis. The Korea National Health and Nutrition Survey (KNHANES) collects data on diseases and nutrition of 7000 people annually, generating national statistics every three years. In its seventh edition of the KNHANES VII survey, oral examinations were performed on the tooth surface in accordance with the World Health Organization (WHO) survey criteria [10], and the number of prosthetic crowns for each participant was determined. Further, the community periodontal index (CPI) was recorded to assess the periodontal status [11]. The present study analyzed the association between the presence of periodontitis and the number of dental prostheses in individuals based on the results of the KNHANES.

## 2. Materials and Methods

This study used data acquired from KNHANES VII, which was a cross-sectional and nationally representative survey conducted by the Korea Centers for Disease Control and Prevention between 2016 and 2018. KNHANES VII is a national cross-sectional survey conducted by the Korean Disease Control and Prevention Agency (KDCA). The survey protocols and secondary use of data were approved by the Institutional Review Board of the KDCA (IRB No. 2018-01-03-P-A). All the study participants provided written informed consent.

The sampling protocol used was a complex, stratified, multistage probability cluster survey of a representative sample of the non-institutionalized civilian population of Korea. A total of 12,689 participants (8708 males and 3981 females), aged 19 years or older, completed the KNHANES VII. Individuals without data on periodontitis, age, and sex were excluded from the analysis. From all the data collected by the KNHANES VII, we used the data on sociodemographic characteristics (age, sex, household income, and level of education), oral health-related variables (toothbrushing, interdental brush use, dental floss use, dental clinic visits, and CPI), oral and systematic medical factor variables (smoking, diabetes mellitus [DM], hypercholesterolemia, hypertension, and body mass index [BMI]), and the number of prosthetic crowns. Pontics were not included as a prosthetic crown.

### 2.1. Periodontal Examination

The oral health component of KNHANES obtained data to assess the oral health status of Koreans, monitor trends in risk factors, and determine the prevalence of dental caries and periodontitis. The oral health data quality control of KNHANES was composed of three parts: ‘Education Program’ and ‘Field Training Program’ for quality control of oral health examiners (dentist) by the professional academy, and ‘Data management’ by CDC. Through these efforts, the data were obtained, and optimum quality level was maintained. All oral examinations of KNHANES were performed by dentists who were highly trained and continuously performed quality control. Periodontal status was evaluated using the CPI probe developed by the WHO [12]. A CPI probe that met the 1997 WHO guidelines [10] was used on 10 index teeth, 2 molars in each posterior sextant, and the upper right and lower left central incisors at 6 sites per tooth (mesiobuccal, buccal, distobuccal, mesiooral, oral, and distooral). The periodontal pocket depth was measured using a CPI probe, distinguished by 3.5 and 5.5 mm. Probing was conducted by dentists trained in calibration. The 5 recorded CPI scores were as follows: CPI 0, healthy and no signs of inflammation; CPI 1, gingival bleeding; CPI 2, presence of gingival calculus; CPI 3, shallow periodontal pocket (>3.5 mm and ≤5.5 mm), and CPI 4, deep periodontal pocket (>5.5 mm). Periodontitis was defined as a CPI score of 3 or 4. Participants were classified into 2 groups: non-periodontitis and periodontitis group.

### 2.2. Number of Crowned Teeth

The number of crowns was counted using the WHO oral health examination criteria [10]. When all surfaces of a tooth were treated, the tooth was determined to a crown. The number of crowns in the anterior and posterior regions was divided, taking into account the clinical characteristics of each region. The anterior teeth were categorized into two groups based on the median number of crowns, and the molars were classified into 1, 2–3, 4–5, or 6 according to the interquartile range.

### 2.3. Covariates

The covariates of this study were the following major sociodemographic factors: sex, age, household income, and education [13]. Household income was classified as <25% (the lowest quartile group), 25–49%, 50–74%, and 75–100% (the highest quartile group). Education level was classified into 4 groups based on the Korean education system: below primary school, middle school, high school, and college or higher education. The oral health variables were toothbrushing, interdental brushing, dental flossing, and dental clinic visits. The systematic medical factor variables were smoking, DM, hypercholesterolemia, hypertension, and BMI. Participants were categorized into 2 groups based on their smoking experience: “non-smoker” and “current or past smoker.” With respect to DM, participants were classified into 3 groups: normal, impaired fasting glucose, and diabetes. The systemic medical factors included in the analysis were DM, hypercholesterolemia, hypertension, and obesity. With respect to DM, participants were classified into 3 groups defined as: a fasting plasma glucose level ≥126 mg/dL, a previous diagnosis of diabetes by a physician, or current use of anti-diabetic agents or insulin [14]. Hypercholesterolemia was defined as a total plasma cholesterol level of ≥240 mg/dL or current use of cholesterol-lowering agents [15]. Hypertension was defined as an average SBP/DBP ≥ 140/90 mmHg or the use of antihypertensive agents [16]. Based on the WHO redefined criteria for obesity in the Asia-Pacific region, obesity was defined as a BMI of ≥25 kg/m^2^ [17].

### 2.4. Statistical Analysis

Data were analyzed using SPSS version 23.0 (SPSS, Chicago, IL, USA). All data were weighted for statistical analyses to account for the complex multistage, stratified, and unequally weighted or clustered sampling design of the KNHANES VII. Appropriate sample weighting factors were selected as specified for each national dataset. Pearson’s chi-square test and independent t-test were used to compare the characteristics of subjects in the periodontitis and non-periodontitis groups. Multivariate logistic regression analyses were used to calculate the adjusted odds ratio (aOR) between the number of prosthetic crown teeth and periodontitis after adjusting for potential confounders. Regression model 1 was adjusted for age and sex. Household income and level of education were added to regression model 2. Oral health variables (toothbrushing, interdental brushing, dental flossing, and dental clinic visits) were added to regression model 3. Systematic medical factor variables (smoking, DM, hypercholesterolemia, hypertension, and BMI) were added to regression model 4. Other multivariate logistic regression analyses were performed to identify the modifiers, such as sex and age, after adjusting for potential confounders in model 4. *p* <0.05 was considered statistically significant.

## 3. Results

### Demographics and Clinical Characterization

Table 1 shows characteristics of the study population in this study, stratified by the number of prosthetic crowns. A total of 12,689 participants were included, of whom 5545 were males, and 7144 were females aged 19 years or older. People with dental prosthetic crowns accounted for 31.3% of the total population.

There were many statistically significant differences in terms of the number of prosthetic crowns among the different socioeconomic status (age, income, and education), personal health practice (use of interdental brush and dental floss, and dental clinic visits), and systematic medical factors (smoking, DM, hypercholesterolemia, hypertension) groups (Table 1). 

Table 2 shows the results of the hierarchical regression analyses that were used to determine the presence of a multivariable association between the number of prosthetic crowns and the prevalence of periodontal disease. The four logistic regression models were designed to adjust hierarchically for covariates. In all the models, individuals with more than six prosthetic crowns in the entire mouth were significantly different from those without any prosthetic crowns. In model 4, compared to respondents with zero prosthetic crowns, respondents with 6–10 and ≥11 crowns were significantly more likely to have periodontitis (aOR = 1.25, 95% confidence interval [CI] = 1.05–1.46 and aOR = 1.28, 95% CI = 1.01–1.62, respectively).

Results of sub-analysis by site revealed that respondents with two or more prosthetic crowns had an aOR of 1.16 (95% CI 1.02–1.32) in the anterior part. In addition, periodontal disease was significantly more likely to occur in respondents with four or more prosthetic crowns in the posterior sites, 1.16 (95% CI 1.00–1.35) was the aOR in respondents with four to five prosthetic crowns, and it was 1.26 (95% CI 1.08–1.47) in respondents with six or more crowns. 

Table 3 shows the results of the logistic regression analyses for multivariable associations between periodontitis and the number of prosthetic crowns after adjusting for demographic variables, socioeconomic variables, oral health variables, and systematic medical factors and stratifying by age and sex. In the female group, the aOR for respondents with 6–10 prosthetic crowns in the entire mouth was 1.26 (95% CI 1.00–1.58), and for respondents with more than 11 crowns, it was 1.63 (95% CI 1.20–2.20) when periodontal disease was the outcome of the model. In the 19–65 years age group, there was a relationship between the number of anterior prosthetic crowns and periodontal disease. In respondents with two or more crowns in the anterior site, the odds of having periodontal disease were 1.27 (95% CI 1.07–1.50). However, in the >65 years age group, there was no significant association between the prevalence of periodontal disease and the number of anterior prosthetic crowns. In the age group of over 65 years, there was a significant association between the prevalence of periodontitis stratified by the number of prosthetic crowns in the posterior site. Respondents with more than six crowns had 1.55 (95% CI 1.20–2.00) times the odds of having periodontal disease, and the respondents with 4–5 prosthetic crowns had 1.42 (95% CI 1.17–1.81) times the odds of having periodontal disease as compared to the respondents with 0–1 prosthetic crown.

## 4. Discussion

The hypothesis that the shape and margin of a dental prosthesis can cause periodontal inflammation has been previously reviewed in several papers [18,19,20,21]. Manufacturing processes from precise impressions to margin formation and final dental prosthesis fabrication are known to be associated with plaque retention and clinical loss of attachment. [22,23,24]. However, thus far, it is understood that if the patient can perform effective plaque control independently and with regular supportive periodontal care, optimal restoration margins within the gingival sulcus do not cause gingival inflammation [9]. 

To prevent disease of teeth with prostheses, more thorough dental care and controlling methods are needed compared to normal teeth [25,26,27,28]. For example, plaque accumulation around the prosthesis and inappropriate oral health management can cause periodontal disease and secondary caries [29,30]. To prevent periodontal disease, various oral care products and special brushing methods are recommended according to the type of prosthesis. 

Table 1 shows the ratio of the presence or absence of periodontal disease according to the number of prostheses. In addition, 18.0% of subjects without periodontal disease had six or more dental prostheses, and over 33.3% of subjects with periodontal disease had six or more dental prostheses. Further, the prevalence of periodontal disease increased as the number of prostheses increased in both anterior and posterior areas (Table 2). 

In particular, as shown in models 3 and 4 of Table 2, despite the adjustment for the covariates of oral health behavior, systemic diseases, and demographic factors, the prevalence of periodontitis increased when the number of prostheses increased in the molar and anterior teeth. Most people with periodontal disease who have prostheses have insufficient self-care for their oral health, therefore, professional oral health management is essential for the progression and prevention of periodontal disease [31,32,33,34].

Results revealed that the disease is more likely to occur in men than in women (Table 1). Periodontal diseases are affected by various factors such as the degree of health concern, sex, hormonal factors, and oral care behavior [35,36,37,38]. The high prevalence of periodontal disease in men could reflect the difference between the aforementioned oral hygiene management ability and interest.

On the other hand, the tendency for the prevalence of periodontal disease to increase with an increase in the number of prosthetic crowns is more evident in women (Table 3). This phenomenon may be explained by a higher tendency among women to receive dental treatment as compared with men, and an increase in prosthetic restorations may make oral hygiene maintenance more difficult. A similar tendency was seen in analyzing prosthetic treatment rates and the prevalence of periodontal disease according to health concerns [39,40].

In the present study, in people under 65 years, the prevalence of periodontal disease was more apparent with an increase in the number of anterior prostheses than with an increase in posterior prostheses (Table 3). On the contrary, in older people, as expected, the higher the number of molar prostheses, the higher the prevalence of periodontal disease.

In adults less than 65 years of age, the presence of anterior tooth prostheses is more likely to be associated with severe dental caries and poor oral hygiene [41,42]; therefore, they have a higher probability of having periodontal disease due to insufficient plaque control. The results in adults aged above 65 years show that the loss of facial muscle function with increasing age is likely to cause food accumulation on the buccal aspect [43,44], thereby making the management of the molar prostheses difficult. In addition, the frequency of use of oral care products tends to decrease with age, accelerating periodontal disease and leading to poor oral hygiene, and factors such as systemic diseases may contribute to fewer visits to the dentist and, thus, affect the occurrence of periodontal disease. 

There are some limitations to consider when interpreting the findings of our study. Firstly, this study has a cross-sectional design; thus, it is not possible to show causality between the occurrence of periodontitis and the number of prosthetic crowns. Future population-based studies that adopt a prospective design are needed. However, cross-sectional studies such as the KNHANES may have a feasible study design, as they do not pose potential ethical issues and difficulty in observing periodontitis. Secondly, the oral hygiene status, such as plaque index, could not be adjusted. However, the usage of toothbrush, floss, and interdental brush were adjusted. Thirdly, there were limitations on the classification criteria of the dependent variable periodontal disease through CPI, and there was a possibility that the partial measurement of periodontal disease may cause bias in [45] the non-measurable point of pontics in the prostheses. It is believed that these areas should be supplemented by further studies using clinical cohort research. In addition, KNANES data utilizing self-report questionnaires may cause recall bias of the participants. 

Despite these limitations, there is an advantage in the use of large-scale national datasets as they can generate results that represent the general population.

## 5. Conclusions

In conclusion, we found that in adults, the number of prosthetic crowns is directly related to the prevalence of periodontitis. Hence, people with a large number of prosthetic crowns require more specialized oral care due to restricted self-management, and this aspect needs to be explored further in future studies.

## Figures and Tables

**Table 1 ijerph-18-05957-t001:** Characteristics of the study population stratified by periodontitis.

Variables	Periodontitis ^1^				*p*-Value
	Normal		Periodontitis ^1^		
	Unweighted N	Weighted % (95% CI)	Unweighted N	Weighted % (95% CI)	
Age (mean ± SD)	8708	46.8 ± 0.3	3981	58.9 ± 0.3	<0.001
Sex					<0.001
Male	3417	37.8 (36.8–38.9)	2128	52.1 (50.4–53.8)	
Female	5291	62.2 (61.1–63.2)	1853	47.9 (46.2–49.6)	
Income					<0.001
Low	1331	15.1 (13.8–16.6)	1009	25 (22.7–27.4)	
Middle low	2011	22.8 (21.4–24.3)	1077	27.2 (25.4–29.1)	
Middle high	2552	29.1 (27.7–30.6)	992	25.1 (23.3–26.9)	
High	2796	33 (30.9–35.1)	888	22.7 (20.5–25.1)	
Education					<0.001
≤Elementary school	1172	13.4 (12.2–14.6)	1147	29.8 (27.4–32.3)	
Middle school	654	7.8 (7.0–8.7)	522	14.7 (13.2–16.3)	
High school	2827	34.7 (33.3–36.3)	1134	30.4 (28.3–32.5)	
≥University or college	3686	44 (42.1–46)	968	25.1 (22.9–27.5)	
Smoking					<0.001
Everyday	1303	14.8 (13.8–15.8)	969	23.8 (22.2–25.5)	
Occasionally	1634	18.3 (17.4–19.2)	988	24.7 (23.1–26.4)	
Never	5699	66.9 (65.7–68.2)	1978	51.5 (49.7–53.3)	
Daily toothbrushing					<0.001
No	89	1 (0.8–1.3)	80	2.1 (1.7–2.7)	
Yes	8545	99 (98.7–99.2)	3855	97.9 (97.3–98.3)	
Flossing ^2^					<0.001
No	6168	71.2 (69.8–72.5)	3368	85.4 (83.8–86.8)	
Yes	2467	28.8 (27.5–30.2)	568	14.6 (13.2–16.2)	
Interdental brushing ^2^					<0.001
No	6833	78.9 (77.8–79.9)	3355	85.9 (84.6–87.2)	
Yes	1802	21.1 (20.1–22.2)	581	14.1 (12.8–15.4)	
Dental clinic visits ^3^					0.082
No	3555	41.1 (39.8–42.4)	1689	43.1 (41.2–45)	
Yes	5078	58.9 (57.6–60.2)	2245	56.9 (55–58.8)	
Diabetes ^4^					<0.001
Normal	5793	70.7 (69.4–71.9)	1859	51.4 (49.4–53.4)	
Impaired fasting glucose	1725	21.2 (20.2–22.2)	1119	28.6 (27–30.3)	
Diabetes	701	8.1 (7.4–8.9)	765	20 (18.5–21.6)	
Hypercholesterolemia ^5^					<0.001
Normal	6584	79.8 (78.7–80.9)	2734	72.8 (71.1–74.3)	
Abnormal	1650	20.2 (19.1–21.3)	1009	27.2 (25.7–28.9)	
Hypertension ^6^					<0.001
Normal	4440	51.5 (50.1–52.9)	1180	29.9 (28.1–31.8)	
Prehypertension	2084	24.1 (23.1–25.2)	1005	25.4 (23.9–27)	
Hypertension	2167	24.3 (23–25.7)	1786	44.6 (42.6–46.7)	
BMI	23.63 ± 0.05		24.47 ± 0.07		<0.001
Number of prosthetic crowns					<0.001
≥11	359	4.1 (3.6–4.7)	342	8.6 (7.6–9.8)	
6–10	1225	13.9 (13.1–14.8)	961	24.7 (23.1–26.3)	
1–5	4041	47.1 (45.9–48.4)	1794	44.7 (43.1–46.4)	
0	3083	34.8 (33.6–36.1)	884	21.9 (20.5–23.4)	

^1^ Periodontitis was defined as community periodontal index codes 3 and 4. ^2^ Daily use of interdental toothbrush and dental floss. ^3^ Dental clinic visits within a year. ^4^ Impaired fasting glucose was defined as 100 mg/dL ≤ fasting blood glucose <126 mg/dL, and diabetes was defined by fasting blood glucose ≥126 mg/dL or current use of anti-diabetic drugs or insulin. ^5^ Hypercholesterolemia was defined by total cholesterol ≥240 mg/dL or current use of drugs for lowering cholesterol. ^6^ Prehypertension was defined as 140 mmHg > systolic blood pressure ≥130 mmHg or 90 mmHg > diastolic blood pressure ≥ 85 mmHg, and hypertension was defined as systolic blood pressure ≥140 mmHg or diastolic blood pressure ≥ 90 mmHg or drug.

**Table 2 ijerph-18-05957-t002:** Multivariable association between the number of prosthetic crowns and periodontitis.

Number of Prosthetic Crowns	Adjusted Odds Ratio (95% Confidence Interval)			
	Model 1	Model 2	Model 3	Model 4
Total	N = 12,689	N = 12,088	N = 12,059	N = 11,406
≥11	1.32 (1.06–1.64)	1.31 (1.05–1.63)	1.33 (1.07–1.67)	1.28 (1.01–1.62)
6–10	1.29 (1.11–1.50)	1.24 (1.06–1.45)	1.28 (1.09–1.51)	1.24 (1.05–1.46)
1–5	1.01 (0.91–1.13)	1.02 (0.92–1.14)	1.06 (0.95–1.19)	1.02 (0.9–1.15)
0	Reference	Reference	Reference	Reference
Anterior				
≥2	1.28 (1.14–1.43)	1.21 (1.07–1.37)	1.22 (1.08–1.38)	1.16 (1.02–1.32)
0–1	Reference	Reference	Reference	Reference
Posterior				
≥6	1.21 (1.05–1.40)	1.22 (1.06–1.41)	1.25 (1.08–1.45)	1.26 (1.08–1.47)
4–5	1.20 (1.05–1.37)	1.14 (0.99–1.32)	1.17 (1.02–1.35)	1.16 (1.00–1.35)
2–3	1.03 (0.91–1.17)	1.05 (0.92–1.19)	1.08 (0.94–1.23)	1.06 (0.93–1.22)
0–1	Reference	Reference	Reference	Reference

Response variable: periodontitis. Explanatory variable: number of prosthetic crowns. Model 1 adjusted for demographic variables (sex and age). Model 2 adjusted for the same factors as model 1 plus socioeconomic variables (household income and level of education). Model 3 adjusted for the same factors as model 2 plus oral health variables (toothbrushing, interdental brushing, dental flossing, and dental clinic visits). Model 4 adjusted for the same factors as model 3 plus systematic medical factor variables (smoking, diabetes mellitus, hypercholesterolemia, hypertension, and body mass index).

**Table 3 ijerph-18-05957-t003:** Multivariable association between the number of prosthetic crowns and periodontitis by sex and age stratification.

Number of Prosthetic Crowns	Adjusted Odds Ratio (95% Confidence Interval)			
	Male	Female	Age < 65 Years	Age ≥ 65 Years
Total	N = 4989	N = 6417	N = 8859	N = 2547
≥11	0.78 (0.55–1.10)	1.63 (1.20–2.20)	1.11 (0.76–1.61)	1.48 (1.02–2.14)
6–10	1.19 (0.91–1.56)	1.26 (1.00–1.58)	1.23 (1.01–1.49)	1.27 (0.92–1.74)
1–5	0.97 (0.82–1.15)	1.05 (0.87–1.26)	0.98 (0.86–1.12)	0.98 (0.71–1.37)
0	Reference	Reference	Reference	Reference
Anterior				
≥2	1.10 (0.92–1.32)	1.21 (1.01–1.46)	1.27 (1.07–1.50)	1.13 (0.94–1.36)
0–1	Reference	Reference	Reference	Reference
Posterior				
≥6	1.02 (0.78–1.34)	1.42 (1.16–1.73)	1.12 (0.91–1.37)	1.55 (1.20–2.00)
4–5	1.05 (0.83–1.32)	1.26 (1.03–1.54)	1.07 (0.89–1.29)	1.42 (1.11–1.81)
2–3	1.01 (0.84–1.22)	1.10 (0.91–1.34)	1.05 (0.90–1.24)	1.11 (0.86–1.44)
0–1	Reference	Reference	Reference	Reference

Response variable: periodontitis. Explanatory variable: number of prosthetic crowns. Models adjusted for demographic variables (sex and age), socioeconomic variables (household income and level of education), oral health variables (toothbrushing, interdental brushing, dental flossing, and dental clinic visits), and systematic medical factor variables (smoking, diabetes mellitus, hypercholesterolemia, hypertension, body mass index).

## Data Availability

The data from the KNHANES VII survey can be accessed and downloaded from the KNHANES homepage (URL: https://knhanes.cdc.go.kr/knhanes/eng/index.do accessed on 29 April 2021).

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
