# Peer review of "Association between the Number of Prosthetic Crowns and Periodontitis: The Korea National Health and Nutrition Examination Survey (KNANES VII) from 2016–2018"

_ijerph, 2021, doi:10.3390/ijerph18115957_

Round 1
Reviewer 1 Report
I thank you for the opportunity to review the manuscript entitled “Association between the Number of Prosthetic Crowns and Periodontitis: A Cross-Sectional Study".
The cross-sectional study addresses a problem that is of considerable importance to dental public health. The study is interesting and novel as currently data on the relationship between crown restorations and periodontitis are scanty.
I deem the methodological framework of the study sound and the reporting in the paper, with some exceptions, is accurate and transparent.
However, I have some questions and would like to point out what I consider minor shortcomings of the report in its current form. In the following, in no particular order, I shall outline my questions and suggestions.
- Title: Consider including some information where the study was carried out. You could, for instance, add: a cross-sectional study in the Republic of Korea
- Title: A pedantic question: Why is “A” capitalized after the colon? Please check with the editorial support team whether the journal style requires this capitalization, which I deem a grammatical error.
- I suggest that you use keywords that are not already included in the title. This increases the searchability and thus visability of the study report in electronic databases.
- Introduction (lines 57): I suggest you dispense with the adjective “credible” as otherwise authors of published reports may take offence.
- I think the clarity of the report would be enhanced if you used consistent nomenclature throughout. For instance, on line 65, you refer to fixed prostheses. While crown are certainly included in this category, I find the term fixed prostheses introduces a level of ambiguity. Let me explain why. I would count a three-unit (tooth supported) bridge as one fixed prostheses. For the purposes of your study, however, one would count such a bridge as two crowns (and exclude the pontic). A long story short, my advice is to dispense with the ambiguous umbrella term fixed prostheses.
- I find it unclear how implant-supported crowns and bridges were classified? Were they included or excluded? On what basis? Were radiographs available during the dental examination? Please make sure that the report is clear and transparent!
- How were data on participants’ oral hygiene habits and dental clinic visits gathered? Were these data based on self-reports? Please make sure that the study report furnished this important piece of information.
- M&M: line 92, provide more detailed information on the assessors who carried out the examination. Number, details on calibration procedure, professional experience, blinding, and so forth.
- M&M: Number of crowned teeth. Again, please provide provide more detailed information on the assessors who carried out the examination. Number, details on calibration procedure, professional experience, blinding, and so forth. Crucially, the study report ought to include information information on blinding of the examiners (or the absence of appropriate blinding). To assess the risk for blinding bias, it is important to know whether or not the counting of crowned teeth was performed without knowledge of the results from the periodontal evaluation. Likewise, it is important to know whether or periodontal evaluation was performed without knowledge of the results from the count of the number of crowned teeth.
- Please consider whether sex or gender is the appropriate term for your study report.
- Please ensure the number of digits after the decimal point is consistent throughout the report.
- Provided that the manuscripts progresses towards publication, please ensure with the editorial support team that Table 2 is shown with all data table rows on a single page.
- Table 1, I do not understand (2) legend unclear “oral health care products were usually used”. Please use wording that is clearer.
- Make sure that, once introduced, abbreviations are consistently used (e.g., BMI).
- You included 12869 participants. According to Table 1, 3966 (3083+884) persons had no crown. I am confused why it is stated on line 144-145 that 31.3% of the population had dental prosthetic crowns. If my back-of-the-envelope calculation is correct, the percentage of persons with no crowns was 30.8% … Please clarify!
- Line 222: Some of the reference you give to support this statement refer to studies investigating implants (24) and edentulous persons (27). Please make sure that you only give pertinent references to reliable sources.
- Line 223: The phrase “Plaque accumulation under the prosthesis” must be revised. It is the biofilm on the surface and at the margin of restorations that may cause problems.
- Line 235-237: Please rephrase this sentence. In its current form, find its meaning unclear.
- Line 258: Your claim that young people often undergo anterior tooth restorations to improve esthetics needs to be supported by robust evidence that is applicable to the population of the Republic of Korea. If no such evidence is available, which I – frankly speaking – please revise said paragraph. Hypotheses and evidence-based statements must be clearly identifiable as what they are.
- Line 264: I find the wording “control oral hygiene independently” unclear. Would perhaps “perform” be more apt?
- Your periodontal classification was based on CPI assessments. Your discussion needs a paragraph detailing the potential for bias. A recent study has shown how partial-mouth measurements affect odds ratios in periodontal research (J Clin Periodontol 2020, Aslhihayb et al.). Please direct the attention of the reader to this important issue.
Author Response
Thank you very much for your detailed review and good comments to improve the completeness of this research.
The authors did their best to revise the description of the study to reflect the opinions of the reviewers.
Review 1)
Title: Consider including some information where the study was carried out. You could, for instance, add: a cross-sectional study in the Republic of Korea
Title: A pedantic question: Why is “A” capitalized after the colon? Please check with the editorial support team whether the journal style requires this capitalization, which I deem a grammatical error.
Answer 1)
Thank you for your considerable review opinion. I edited the title reflecting your review opinion (Ln 2-3)
Review 2)
I suggest that you use keywords that are not already included in the title. This increases the searchability and thus visability of the study report in electronic databases.
Answer 2)
The title has been revised to reflect the review comments. (Ln2-3)
Review 3)
Introduction (lines 57): I suggest you dispense with the adjective “credible” as otherwise authors of published reports may take offence.
Answer 3)
It has been deleted to reflect the review opinion. (ln 57)
Review 4)
I think the clarity of the report would be enhanced if you used consistent nomenclature throughout. For instance, on line 65, you refer to fixed prostheses. While crown are certainly included in this category, I find the term fixed prostheses introduces a level of ambiguity. Let me explain why. I would count a three-unit (tooth supported) bridge as one fixed prostheses. For the purposes of your study, however, one would count such a bridge as two crowns (and exclude the pontic). A long story short, my advice is to dispense with the ambiguous umbrella term fixed prostheses.
Answer 4)
The term has been changed ‘fixed’ prostheses to ‘dental’ prostheses to reduce confusion about by reflecting the review opinions. (Ln 66)
Review 5)
I find it unclear how implant-supported crowns and bridges were classified? Were they included or excluded? On what basis? Were radiographs available during the dental examination? Please make sure that the report is clear and transparent!
Answer 5)
In order to reinforce the contents by reflecting the review opinions, the oral examination part has added a description of the oral examination method of the National Health and Nutrition Examination Survey operated in Korea. (Ln88-95)
As an additional explanation, the oral examination method used in Korea borrows the oral examination standards of WHO oral examination protocol. The dental prosthesis is inspected by a trained dentist and a questionnaire for the patient is performed. In the case of an implant crown and a pontics, it is a prosthesis restored in the place of the extracted tooth, and according to the WHO oral examination standard, it is recorded as a caries or inexperienced restored tooth. It is not possible to classified among dental restorations without caries experiences.
Review 6)
How were data on participants’ oral hygiene habits and dental clinic visits gathered? Were these data based on self-reports? Please make sure that the study report furnished this important piece of information.
Answer 6)
The data was described by adding the relevant content to the limiting part of the discussion about the bias that may be caused by the survey by the subject's direct questionnaire response (Ln286-287)
Review 7)
M&M: line 92, provide more detailed information on the assessors who carried out the examination. Number, details on calibration procedure, professional experience, blinding, and so forth.
M&M: Number of crowned teeth. Again, please provide provide more detailed information on the assessors who carried out the examination. Number, details on calibration procedure, professional experience, blinding, and so forth. Crucially, the study report ought to include information information on blinding of the examiners (or the absence of appropriate blinding). To assess the risk for blinding bias, it is important to know whether or not the counting of crowned teeth was performed without knowledge of the results from the periodontal evaluation. Likewise, it is important to know whether or periodontal evaluation was performed without knowledge of the results from the count of the number of crowned teeth.
Answer 7)
In order to reinforce the contents by reflecting the review opinions, the oral examination part has added a description of the oral examination method of the National Health and Nutrition Examination Survey operated in Korea. (Ln88-95)
Review 8)
Please consider whether sex or gender is the appropriate term for your study report.
Answer 8)
From a scientific point of view, the term ‘sex’ was used rather than the use of the term gender’, giving more meaning to biological factors than sociological factors.
Review 9)
Please ensure the number of digits after the decimal point is consistent throughout the report.
Answer 9)
Reflecting the review opinion, the number of decimal places at the same level has been unified. (Table)
Review 10)
Provided that the manuscripts progresses towards publication, please ensure with the editorial support team that Table 2 is shown with all data table rows on a single page.
Answer 10)
I edited it as a draft. We will finalize and revise the contents before publication. (Table)
Review 11)
Table 1, I do not understand (2) legend unclear “oral health care products were usually used”. Please use wording that is clearer.
Answer 11)
It has been revised to check whether oral care products are used or not by reflecting the review opinions. (Ln156-163)
Review 12)
Make sure that, once introduced, abbreviations are consistently used (e.g., BMI).
Answer 12)
Except for the legend of the table, it was confirmed that the abbreviation was used continuously after mentioning it once.
Review 13)
You included 12869 participants. According to Table 1, 3966 (3083+884) persons had no crown. I am confused why it is stated on line 144-145 that 31.3% of the population had dental prosthetic crowns. If my back-of-the-envelope calculation is correct, the percentage of persons with no crowns was 30.8% … Please clarify!
Answer 13)
This study is the complex multistage, stratified, and unequally weighted or clustered sampling design sample study, and the 31.3% of the number described in the text is not a simple arithmetic calculation, but a calculated percentage with weights included. Thank you for your review comments.
Review 14)
Line 222: Some of the reference you give to support this statement refer to studies investigating implants (24) and edentulous persons (27). Please make sure that you only give pertinent references to reliable sources.
Answer 14)
In reflection of the review opinion, reference number 24 was excluded as it was deemed inappropriate as a study for patients with fixed implant prosthesis.
Review 15)
Line 223: The phrase “Plaque accumulation under the prosthesis” must be revised. It is the biofilm on the surface and at the margin of restorations that may cause problems.
Answer 15)
In order to prevent confusion in context by reflecting the review opinion, it has been changed to the expression of ‘around’ prosthesis. (Ln 234)
Review 16)
Line 235-237: Please rephrase this sentence. In its current form, find its meaning unclear.
Answer 16)
It has been modified to reflect the review opinions. Most people with periodontal disease who have prostheses have insufficient self-care for their oral health, therefore professional oral health management is essential for the progression and prevention of periodontal disease. (Ln246-248)
Review 17)
Line 258: Your claim that young people often undergo anterior tooth restorations to improve esthetics needs to be supported by robust evidence that is applicable to the population of the Republic of Korea. If no such evidence is available, which I – frankly speaking – please revise said paragraph. Hypotheses and evidence-based statements must be clearly identifiable as what they are.
Answer 17)
Reflecting the review opinion, it was difficult to add epidemiological evidence, so the corresponding phrase was deleted.
Review 18)
Line 264: I find the wording “control oral hygiene independently” unclear. Would perhaps “perform” be more apt?
Your periodontal classification was based on CPI assessments. Your discussion needs a paragraph detailing the potential for bias. A recent study has shown how partial-mouth measurements affect odds ratios in periodontal research (J Clin Periodontol 2020, Aslhihayb et al.). Please direct the attention of the reader to this important issue.
Answer 18)
In reflection of the review opinion, a description of the possibility of bias occurrence according to the measurement of partial periodontal disease in Aslhihayb et al’s research was added to the limitation part of the discussion part.

Reviewer 2 Report
dear Authors,
congratulations for the study.
Please check the spelling of some words.
regarding the content I would discuss better the role of the prosthetic margin in the manufacturing process and the role of an accurate impression in the realization of an accurate margin
(https://pubmed.ncbi.nlm.nih.gov/33435664/, https://pubmed.ncbi.nlm.nih.gov/25438741/)
Author Response
Thank you very much for your detailed review and good comments to improve the completeness of this research.
The authors did their best to revise the description of the study to reflect the opinions of the reviewers.
Review)
Please check the spelling of some words.
regarding the content I would discuss better the role of the prosthetic margin in the manufacturing process and the role of an accurate impression in the realization of an accurate margin
(https://pubmed.ncbi.nlm.nih.gov/33435664/, https://pubmed.ncbi.nlm.nih.gov/25438741/)
Answer)
References were added to the discussion part by reflecting the relevant content, and the final shape was calculated according to the prosthesis manufacturing process and the contents of the plaque accumulation were revised.
